

# 24-hour movement behaviours and self-rated health in Chinese adolescents: a questionnaire-based survey in Eastern China

Guanghui Shi[1], Chaomei Liang[2], Wanli Zang[3], Ran Bao[4,5], Jin Yan[4,5], Li Zhou[6] and Lei Wang[7]

[1] Ningbotech University, Department of Physical Education, Zhejiang, China
[2] Library of Beijing Sport University, Beijing, China
[3] Postgraduate School, Harbin Sport University, Harbin, China
[4] Centre for Active Living and Learning, University of Newcastle, Callaghan, NSW, Australia
[5] College of Human and Social Futures, University of Newcastle, Callaghan, NSW, Australia
[6] School of Sports and Health, Guizhou Medical University, Guiyang, China
[7] School of Physical Education, Shanghai University of Sport, Shanghai, China

Corresponding author
Li Zhou, zhouli@gmc.edu.cn

## ABSTRACT

**Objective.** Although much evidence has demonstrated the benefits of adhering to the 24-hour movement guidelines, little is known about their association with self-rated health in adolescents. The objective of this study was to explore the association between adherence to the 24-hour movement guidelines (*i.e.*, physical activity, screen time, and sleep) and self-rated health among Chinese adolescents aged 10–17 ages.

**Methods.** A convenient sample of 12 schools and their students were recruited for this cross-sectional study. Physical activity and screen time were assessed using a questionnaire based on the Health Behaviour in School-aged Children questionnaire, while sleep duration was used measured using the Pittsburgh Sleep Scale. Ordinal logistic regression was performed to examine the association between adherence to the 24-hour movement guidelines and self-rated health.

**Results.** Adolescents who adhered to more guidelines contained in the 24-hour movement guidelines reported better self-rated health. A dose-response association was observed, with the odds ratio (OR) for adhering to all three = 11.26, 95% CI [4.82–26.35]; OR for meeting two = 1.62, 95% CI [1.25–2.12]; OR for meeting one = 1.16, 95% CI [0.98–1.38]) indicating a higher probability of better self-rated health with increasing adherence. Regarding adherence to specific combination of 24-hour movement guidelines, positive associations were found for adhering to all three recommendations (OR = 11.95, 95% CI [5.06–28.19]), only MVPA (OR = 4.96, 95% CI [2.82–8.72]), MVPA + screen time (OR = 5.50, 95% CI [3.02–9.99]), and MVPA + sleep (OR = 4.63, 95% CI [2.52–8.51]).

**Conclusion.** This study provides evidence supporting the association between adherence to the 24-hour movement guidelines and better self-rated health among Chinese adolescents. Sufficient physical activity may be particularly important for promoting self-rated health in this population.

## INTRODUCTION

Since the Canadian 24-hour movement guidelines were issued in 2016 (*Tremblay et al., 2016*; *Liu et al., 2022*; *Ma et al., 2022*), emerging evidence has focused on the benefits of adhering to the movement guidelines. For optimal health outcomes, adolescents should adhere to the recommendation of a higher level of physical activity (*i.e.,* at least 60 min of moderate-to-vigorous physical activity (MVPA) per day) (*Ren, Yan & Sun, 2021*; *Li et al., 2023*; *Liu et al., 2023*), limited screen time (ST) (*i.e.,* no more than 2 h per day) (*Huang et al., 2021*), and sufficient sleep (*i.e.,* 8–10 h per day for 14–17 years) (*Tremblay et al., 2016*; *Liu et al., 2022*; *Li et al., 2023*; *Yan et al., 2023*). Adherence to the 24-hour movement guidelines (*Tremblay, Carson & Chaput, 2016*; *Chen et al., 2022b*) is important for promoting health in children and adolescents (*Sampsa-Kanyinga et al., 2021*). Numerous studies have investigated the association between adherence to the 24-hour movement guidelines and various health indicators (*Ma et al., 2022*; *Rollo, Antsygina & Tremblay, 2020*). For instance, a study of representative samples indicated that adherence to the 24-hour movement guidelines was linked with a lower risk of being overweight and obese in Chinese children and adolescents (*Chen et al., 2021*). *Vanderloo et al. (2021)* found that adhering to the 24-hour movement guidelines was associated with favorable cholesterol outcomes. A recent study of Japanese children suggested that adherence to the 24-hour movement guidelines was linked to higher levels of physical fitness (*Tanaka et al., 2020*). In addition to the association between adherence to the 24-hour movement guidelines and physical health outcomes, a growing body of evidence suggested that adherence to the 24-hour movement was also associated with favorable mental health outcomes (*Liu et al., 2023*; *Sampsa-Kanyinga et al., 2020*). Furthermore, there was evidence suggesting that adherence to the 24-hour movement guidelines was associated with improved academic performance in school children (*Watson et al., 2021*; *Tapia-Serrano et al., 2021*).

Self-rated health (SRH) is a widely used measure in public health to assess an individual's overall health (*Liu et al., 2023*; *Jylhä, 2009*; *Shi et al., 2022*). It refers to the perception of one's health status (*Kastelic et al., 2021*) and serves as an important indicator of an individual's general well-being, reflecting not only their physical health but also their mental, emotional, and social health (*Nieminen et al., 2010*; *Vie et al., 2019*). Evidence has shown that SRH was a strong predictor of future health outcomes, such as all-cause of mortality and morbidity (*Mavaddat et al., 2011*). Adolescence is a crucial period of development, and understanding adolescents' SRH can help identify potential health concerns that may affect their future health. Recently, emerging evidence has indicated that physical activity, screen time, and sleep duration were associated with self-rated health in adolescents (*Zhang, Lu & Wu, 2020*; *Yeo et al., 2019*; *Zheng et al., 2023*). Moreover, adhering to the 24-hour movement guidelines has been linked to better self-rated health among adolescents (*Sampsa-Kanyinga et al., 2022*; *Kyan, Takakura & Miyagi, 2022*). Although previous studies have explored the association between meeting the 24-hour movement guidelines and SRH (*Kyan, Takakura & Miyagi, 2022*; *Vanderloo et al., 2021*), such associations might not be replicated in other contexts (*i.e.,* country). To verify the

effects of healthy movement behaviours on health outcomes, it is necessary to further assess the association between meeting the 24-hour movement guidelines and SRH in other countries, such as China. It has been documented that PA, ST and sleep duration are independent factors of SRH in Chinese adolescents (*Chen et al., 2022b*; *Shi et al., 2022*). However, little is known about their integrative association (*i.e.,* in combination with PA, ST and sleep duration) with SRH. Therefore, it is necessary to explore the associations, which can better inform researchers and policymakers to conduct effective plans to improve SRH.

Although there is accumulating evidence confirming the health benefits of adhering to the 24-hour movement guidelines in children and adolescents, most of this evidence was based on Western populations (*Rollo, Antsygina & Tremblay, 2020*). However, such evidence is scarce in the Chinese population, with only five studies conducted on Chinese children and adolescent samples (*Chen et al., 2021*; *Lu et al., 2021*; *Shi et al., 2020*). This highlights the need for more China-based studies to confirm the health benefits of adhering to the 24-hour movement guidelines and provide evidence for updating and refining the guidelines. Moreover, most previous studies have focused on specific health indicators, such as overweight and obesity, depressive symptoms, and cholesterol levels, rather than an overall health indicator. SRH is a simple and effective measure to assess health status and has been linked with various health outcomes (*Williams, Nardo & Verma, 2017*; *Lorem et al., 2020*; *Gallagher et al., 2016*). Despite this, limited evidence exists regarding the association between adherence to the 24-hour movement guidelines and SRH in Chinese children and adolescents (*Sampasa-Kanyinga et al., 2021*; *Sampasa-Kanyinga et al., 2022*; *Kyan, Takakura & Miyagi, 2022*). Therefore, more studies are needed to address this gap in the literature. The current study aimed to explore the association between adherence to the 24-hour movement guidelines and self-rated overall health in a sample of Chinese children and adolescents.

## METHODS

### Participants

This cross-sectional study was conducted in Shanghai, Anhui Province, and Jiangsu Province using a convenience sampling method. Research investigators contacted schools in different regions and invited 3,807 adolescents from 12 senior high schools to participate, of whom 2,407 students returned fully completed questionnaires containing the information for variables of interest. Before commencing data collection, the school leaders of each involved school were informed for the purpose of this study. Two researchers closely supervised the data collection process, with assistance from teachers in all the participating schools. Teachers have undergone training to understand the purpose of this study, and to offer explanations in response to student's queries. All guardians or parents provided written informed consent for their children's participation in this study. Additionally, we have obtained informed assent from participants of the current study. This study was approved by the Research Board at Shanghai University of Sport (Approval Number: 102772021RT071). To avoid central ethical issues, all the research staff involved in data

collection, data analysis, and manuscript write-up received comprehensive ethic training. In these regards, the processes of this study were conducted ethically, with no considerations of unethical matters.

The unfilled questionnaire in our study was defined as the questionnaire that the respondents did not provide valid data for the variable of interest this study needed. In this case, those questionnaires were excluded. Generally healthy and literate school-aged students, with no medical history and conditions, were targeted as potential sampling participants; conversely, school-aged students who were not able to fully understand the questionnaire and whose daily behaviours were affected by medical conditions were excluded from the sampling participants.

## Measures
### Exposures
According to the Canadian 24-Hour Movement Guidelines (*Tremblay et al., 2016*), participants who reported engaging in PA for 7 days and having less than 2 h of screen time per day were considered as adhering to the PA and ST recommendations, respectively. For participants aged between 10–12 years and 13–17 years, those who reported having 9–11 h and 8–10 h of sleep duration at night, respectively, were considered as adhering to the sleep recommendations. Adherence to 24-hour movement guidelines was categorized into eight groups: (1) not adhering to any recommendations, (2) adhering to only MVPA recommendation, (3) adhering to only screen time recommendation, (4) adhering to only sleep recommendation, (5) adhering to MVPA and screen time recommendations, (6) adhering to MVPA and sleep recommendations, (7) adhering to screen time and sleep recommendations, (8) adhering to all three recommendations.

The PA and ST data were collected using the Chinese version of the Health Behaviour in School-Aged Children survey questionnaire (*Su et al., 2022*), which has been used in the previous studies in Chinese children and adolescents (*Liu et al., 2010*; *Li et al., 2008*). To measure PA, participants were required to answer the question: "During the past 7 days, how many days did you engage in moderate-to-vigorous physical activity for at least 60 min per day?" Adherence to the MVPA recommendation was indicated by reporting physical activity for 7 days. Screen time was assessed by the question: "On both weekdays and weekends, how many hours did you typically spend doing the following activities in your leisure time, including watching TV or movies or DVDs, playing video games, and engaging in other screen-based activities?" Participants adhered to the screen time recommendation if they reported no more than 2 h of screen time per day. This questionnaire has been validated as a reliable measure of physical activity (intraclass correlation coefficients: 0.82) and screen time (intraclass correlation coefficients for different screen-time measures ranged from 0.33 to 0.74) in Chinese children and adolescents (*Liu et al., 2010*).

The sleep duration was evaluated through the Pittsburgh Sleep Scale using the questions: "During the past month, how many hours did you usually sleep at night (the time when you fell asleep)?" This scale has been widely used in Chinese children and adolescents (*Chen et al., 2022b*; *Chen et al., 2022a*), and also demonstrated good reliability (intraclass correlation coefficient: 0.90) and construct validity (Pearson correlation coefficients compared with

the Center for Epidemiologic Studies Depression Scale for Children and Fatigue Scale—Adolescent ranged from 0.27 to 0.64) in Chinese children and adolescents (*Ho et al., 2021*).

### Outcome

Self-rated health was assessed using a single question "In general, how do you consider your life satisfaction?" Response options were "very good", "good", "fair", or "poor". This question has been associated with good psychometric performance in children and adolescents (*Shi et al., 2022*; *Boardman, 2006*).

### Covariates

Participants self-reported sex (male or female), age, grade (5, 6, 7,…10), siblings (yes or no), living with parent (yes or no), residence (urban, suburban or rural), father educational level (Middle school or below, High school/occupational school, University/College, Master or above, Unknown), mother educational level (Middle school or below, High school/occupational school, University/College, Master or above, Unknown).

### Statistical analysis

The sample size of our study was calculated using G*Power 3.1 software. Data from previous studies were used to select the parameters for the calculation, including the effect size of 3.9 (odds ratio) (*Kyan, Takakura & Miyagi, 2022*), $\alpha$ at 0.05, and power of 95%, which showed that a sample size of 230 was required per group. The sample characteristics were reported using descriptive analysis. To examine the association between adherence to the 24-hour movement guidelines and self-rated health, ordinal logistic regression was performed while controlling for several covariates including sex, age, grade, siblings, living with a parent, residence, father's educational level, and mother's educational level. The reference group for the analysis was set as those who did not adhere to any recommendations of the 24-hour movement guidelines and the probability of reporting low levels of self-rated health was estimated for participants who adhered to more guidelines. All variables were entered into the statistical model, and odds ratios (ORs) with 95% confidence intervals (CI) were presented as the results. Statistical significance was set up as $p < 0.05$. All analyses were performed using SPSS software version 28.

## RESULTS

The descriptive characteristics of participants were outlined in Table 1. In total, 2407 adolescents had valid data and were included in the final analysis, and 46% of included participants were girls. Almost half of the participants have siblings, and 84% of adolescents lived with their parents. The highest proportion of the sample was 16 years old adolescents, and 21.5% of included participants were in 9th grade. In addition, the majority of parents' educational level (>80%) was university/college or below. More than two-thirds of the adolescents lived in urban areas. Regarding movement behaviours, the prevalence of adhering to MVPA guidelines was only 6.4%, and 43.5% of participants adhered to the screen time guidelines, while only 30.3% of adolescents adhered to the sleep guidelines. Moreover, only 0.9% of participants adhered to all three types of movement behaviour
**Table 1   Descriptive characteristics of the participants.**

|  |  | n | % |
|---|---|---|---|
| Sex |  |  |  |
|  | Male | 1,280 | 53.2 |
|  | Female | 1,127 | 46.8 |
| Age |  |  |  |
|  | 10 | 103 | 4.3 |
|  | 11 | 273 | 11.3 |
|  | 12 | 378 | 15.7 |
|  | 13 | 385 | 16 |
|  | 14 | 290 | 12 |
|  | 15 | 179 | 7.4 |
|  | 16 | 614 | 25.5 |
|  | 17 | 185 | 7.7 |
| Grade |  |  |  |
|  | 5 | 353 | 14.7 |
|  | 6 | 333 | 13.8 |
|  | 7 | 350 | 14.5 |
|  | 8 | 417 | 17.3 |
|  | 9 | 517 | 21.5 |
|  | 10 | 437 | 18.2 |
| Siblings |  |  |  |
|  | Yes | 1,185 | 49.2 |
|  | No | 1,222 | 50.8 |
| Live with parent |  |  |  |
|  | Yes | 2,023 | 84 |
|  | No | 384 | 16 |
| Residence |  |  |  |
|  | Rural | 279 | 11.6 |
|  | Suburban | 530 | 22 |
|  | Urban | 1,598 | 66.4 |
| Father educational level |  |  |  |
|  | Middle school or below | 621 | 25.8 |
|  | High school/occupational school | 591 | 24.6 |
|  | University/college | 773 | 32.1 |
|  | Master or above | 139 | 5.8 |
|  | Unknown | 283 | 11.8 |
| Mother educational level |  |  |  |
|  | Middle school or below | 767 | 31.9 |
|  | High school/occupational school | 516 | 21.4 |
|  | University/college | 732 | 30.4 |
|  | Master or above | 114 | 4.7 |
|  | Unknown | 278 | 11.5 |

**Table 1** (*continued*)

|  |  | *n* | % |
|---|---|---|---|
| MVPA guideline |  |  |  |
|  | Not meet | 2,252 | 93.6 |
|  | Meet | 155 | 6.4 |
| Screen guideline |  |  |  |
|  | Not meet | 1,361 | 56.5 |
|  | Meet | 1046 | 43.5 |
| Sleep guideline |  |  |  |
|  | Not meet | 1,678 | 69.7 |
|  | Meet | 729 | 30.3 |
| Combinations of the guidelines met |  |  |  |
|  | None | 867 | 36 |
|  | MVPA only | 43 | 1.8 |
|  | Screen only | 722 | 30 |
|  | Sleep only | 407 | 16.9 |
|  | MVPA & screen | 46 | 1.9 |
|  | MVPA & sleep | 44 | 1.8 |
|  | Screen & sleep | 256 | 10.6 |
|  | All | 22 | 0.9 |
| Number of the guidelines met |  |  |  |
|  | None | 867 | 36 |
|  | One | 1172 | 48.7 |
|  | Two | 346 | 14.4 |
|  | Three | 22 | 0.9 |
| Self-rated health |  |  |  |
|  | Poor | 153 | 6.4 |
|  | Fair | 864 | 35.9 |
|  | Good | 740 | 30.7 |
|  | Very good | 370 | 15.4 |
|  | Excellent | 280 | 11.6 |

guidelines, and 36% of the sample did not adhere to any movement behaviour guidelines. Almost half of the participants adhered to only one type of movement behaviour guidelines. Furthermore, more than 65% of adolescents self-reported their health as "Good" or "Fair", while the prevalence of "Excellent" and "Very good" self-rated health were 11.6% and 15.4%, respectively.

The association between the number of 24-hour movement guidelines adhered to and self-rated health was shown in Table 2. Adherence to two (OR = 1.62, 95% CI [1.25–2.12]) and all (OR = 1.26, 95% CI [4.82–26.35]) types of 24-hour movement guidelines were all positively associated with self-rated health compared to adhering to none of the guidelines. Table 3 presented the association between adherence to the 24-hour movement guidelines (specific combination) and self-rated health. Compared to adhering to none of the 24-hour movement guidelines, adherence to all guidelines (OR = 11.95, 95% CI [5.06–28.19]), only MVPA guidelines (OR = 4.96, 95% CI [2.82–8.72]), MVPA + screen time guidelines

**Table 2  Association between adherence to the 24-hour movement guidelines and self-rated health.**

|  | OR | 95% CI | |
| --- | --- | --- | --- |
| Meeting all | 11.26 | 4.82 | 26.35 |
| Meeting two | 1.62 | 1.25 | 2.12 |
| Meeting one | 1.16 | 0.98 | 1.38 |
| Meeting none | | Reference group | |

**Notes.**
OR, odd ratio; CI, confidence interval; None, meeting any one guidelines of the 24-hour movement guidelines; Two, meeting any two guidelines of the 24-hour movement guidelines; All, meeting all the 24-hour movement guidelines.

**Table 3  Association between adherence to the 24-hour movement guidelines (specific combination) and self-rated health.**

|  | OR | 95%CI | |
| --- | --- | --- | --- |
| Meeting all | 11.95 | 5.06 | 28.19 |
| Meeting MVPA only | 4.96 | 2.82 | 8.72 |
| Meeting screen only | 1.06 | 0.88 | 1.28 |
| Meeting sleep only | 1.17 | 0.89 | 1.53 |
| Meeting MVPA + screen | 5.50 | 3.02 | 9.99 |
| Meeting MVPA + sleep | 4.63 | 2.52 | 8.51 |
| Meeting screen + sleep | 1.14 | 0.84 | 1.55 |
| Meeting none | | Reference group | |

(OR = 5.50, 95% CI [3.02–9.99]), and MVPA + sleep guidelines (OR = 4.63, 95% CI [2.52–8.51]) were positively associated with self-rated health. However, adherence to only screen time guidelines (OR = 1.06, 95% CI [0.88–1.28]), only sleep guidelines (OR = 1.17, 95% CI [0.89–1.53]), and screen + sleep guidelines (OR = 1.14, 95% CI [0.84–1.55]) were not associated with self-rated health.

## DISCUSSION

This study aimed to examine the association between adherence to the 24-hour movement guidelines and self-rated health among Chinese adolescents. A cross-sectional study was conducted, comprising Chinese adolescents from three cities. Results showed that adherence to the 24-hour movement guidelines, along with different combinations of recommendations, was associated with better self-rated health. Specifically, adolescents who adhered to the three recommendations were more likely to report better self-rated health. Notably, adherence to the MVPA recommendation exhibited a stronger association with self-rated health in Chinese adolescents. Moreover, a dose–response association was observed between the number of recommendations adhered to and self-rated health, whereby adolescents who adhered to more recommendations were more likely to report greater self-rated health. These findings underscore the significance of adhering to 24-hour movement guidelines for improving self-rated health among Chinese adolescents.

This study demonstrates that adhering to all three recommendations of the integrated guidelines is more likely to lead to better self-rated health in adolescents compared with adhering to none of the 24-hour movement guidelines. This finding is consistent

with previous research that has shown a positive association between adherence to the 24-hour movement guidelines, including sufficient physical activity, limited screen time, adequate sleep duration, and greater self-rated health (*Kyan, Takakura & Miyagi, 2022*; *Sampasa-Kanyinga et al., 2022*). For instance, the Ontario Student Drug Use and Health Survey conducted in Canada, which examined 5,739 students in 2017 and 6,960 students in 2019, reported that adherence to either individual or combined recommendations contained in the 24-hour movement guidelines was positively associated with self-rated physical and mental health in adolescents (*Sampasa-Kanyinga et al., 2022*). Similarly, a cross-sectional study conducted in Japan involving 4,360 adolescents found that adherence to screen time and sleep recommendations was solely associated with good self-rated health in children (*Kyan, Takakura & Miyagi, 2022*). Moreover, adolescents who adhered to all three recommendations, as well as only MVPA, only sleep, ST and sleep, MVPA and sleep were more likely to report better self-rated health (*Kyan, Takakura & Miyagi, 2022*). These findings are supported by review studies, which indicated that physical activity, sedentary time, and sleep duration in combination have benefits on adolescents' mental health and physical health (*Sampasa-Kanyinga et al., 2020*; *Rollo, Antsygina & Tremblay, 2020*). For instance, the first review study regarding the benefits of adhering to 24-hour movement guidelines found a desirable association between adhering to 24-hour movement guidelines and health indicators in children and adolescents (*Rollo, Antsygina & Tremblay, 2020*). As a critical development period, it is necessary to improve the prevalence of adhering to 24-hour movement guidelines to achieve better health status among children and adolescents. Thus, existing research suggests that considering the combination of physical activity, screen time, and sleep duration has benefits for adolescents' perception of their overall health.

This study also indicates adhering to screen time guidelines and sleep guidelines, either individually or in combination, is not associated with greater self-rated health in adolescents compared with adhering to none of the 24-hour movement guidelines, which is not aligned with previous studies (*Kyan, Takakura & Miyagi, 2022*; *Sampasa-Kanyinga et al., 2022*). Notably, maintaining adequate sleep is critical for promoting self-rated health among adolescents (*Kyan, Takakura & Miyagi, 2022*). Scientific evidence has demonstrated the detrimental effects of insufficient sleep on various academic-related outcomes (*e.g.*, cognition, emotional regulation, and academic achievement) (*Tarokh, Saletin & Carskadon, 2016*) and mental disorders (*e.g.*, depression, anxiety) (*Gregory & Sadeh, 2016*), as well physical health (*e.g.*, adiposity, cardiometabolic biomarkers) (*Chaput et al., 2016*). Therefore, these findings underscore the significance of sleep duration as a significant movement behaviour in maintaining better self-rated overall health among adolescents. Additionally, although this study found that adherence to the screen time guidelines was not associated with self-rated health in adolescents, excessive screen time has been shown to have negative effects on a range of physical health outcomes. These negative effects include insufficient physical activity and sleep duration, increased risk factors for cardiovascular diseases such as obesity, and high blood pressure, as well as poor social and emotional behaviours, depression, anxiety, and cognitive outcomes such as cognitive control and emotional regulation in youths (*Lissak, 2018*). Adhering to screen time recommendations may reduce the physical and mental health risk factors in adolescents

and improve their overall self-perception of health. The insignificant association between adherence to screen time recommendations and self-rated health may attribute to the non-representative sample size. A large and more diverse sample is necessary to accurately to examine the association between adherence to screen time and sleep recommendations and self-rated health in Chinese adolescents. This because non-representative sample size may dilute the association between different movement behaviours and self-rated health. Therefore, using a representative sample is crucial to ensure the generalisability of the results. In contrast to the findings reported in a previous study of Japanese children and adolescents (*Kyan, Takakura & Miyagi, 2022*), our results suggest that adhering to physical activity recommendations, either in isolation or combination with other movement behaviours, was positively associated with better self-rated health in Chinese adolescents. Similarly, one cross-sectional study suggested the strongest association between physical activity and self-rated health among Chinese adolescents (*Yiting et al., 2023*). This suggests that MVPA may be more crucial for self-rated health than other recommendations in Chinese adolescents.

This study found a dose-dependent association between adherence to 24-hour movement guidelines and self-rated health in adolescents. Specifically, adhering to more recommendations contained in the 24-hour movement guidelines was incrementally associated with greater self-rated health in adolescents. This finding is consistent with previous studies that reported a positive association between adherence to more recommendations contained in the 24-hour movement guidelines and greater self-rated health (*Zhang, Lu & Wu, 2020*; *Sampasa-Kanyinga et al., 2022*; *Sun, Jiang & Wei, 2023*). A meta-analysis has also indicated a strong dose–response (OR = 1.57, 95% CI [1.15–2.13], $p = 0.008$) association between higher physical activity levels (moderate PA *versus* low PA) and greater self-rated health in both boys and girls (*Zhang, Lu & Wu, 2020*). The same meta-analysis also confirmed a significant dose–response association between increased screen time (OR = 1.25, 95% CI [1.06–0.47], $p = 0.010$) and poor self-rated health in adolescents. In a cross-sectional study, the combined association between physical activity, screen time and sleep, and self-rated health was explored, revealing a dose–response association for self-rated well-being among children and adolescents (*Sun, Jiang & Wei, 2023*). The results suggested that physical activity, screen time, and sleep duration are independently associated with self-rated health in adolescents, and their combined accumulation leads to improvements in self-rated health. Accordingly, physical activity, screen time, and sleep duration are jointly associated with various health outcomes, which can in turn improve individuals' perceptions of overall health (*Grgic et al., 2018*). Overall, the current study confirms the positive association between adherence to 24-hour movement guidelines and self-rated health and found a stronger association between adherence to MVPA recommendations and self-rated health in Chinese adolescents.

This study has several limitations that should be acknowledged. Firstly, cross-sectional studies can provide valuable insights into the association between movement behaviours, such as physical activity, screen time, and sleep duration, and health outcomes through large population-based surveys. However, they preclude the examination of potential causal effects of adherence to the 24-hour movement guidelines on self-rated health.

Therefore, it is encouraged that longitudinal and intervention study designs should be used to gain a better understanding of the benefits of adhering to the 24-hour movement guidelines. Secondly, self-reported measures are widely used in cross-sectional studies for their cost-effectiveness and ease of administration. Although self-reported measures may suffer from social desirability and recall bias, they have good reliability and validity to capture the type and context of movement behaviours (*Sampasa-Kanyinga et al., 2022*). However, self-reported measures do not assess the amount and intensity of movement behaviours, which may affect the accuracy of reporting physical activity, screen time, and sleep duration. Therefore, further studies are encouraged to combine self-reported measures with device-based measures. Finally, the use of a non-probabilistic sampling strategy may limit the generalizability of our conclusions. Hence, future studies are encouraged to use representative samples to explore the association between adherence to the 24-hour movement guidelines and self-rated health among adolescents.

## CONCLUSION

The findings of this study suggest that adherence to the 24-hour movement guidelines is positively associated with higher levels of self-rated health in adolescents. Specifically, adherence to the MVPA recommendation demonstrates a stronger association with self-rated health in Chinese adolescents. Additionally, a dose–response relationship between adherence to the 24-hour movement guidelines and better self-rated health is observed in Chinese adolescents. These findings highlight the significance of promoting physical activity, appropriate sleep, and limiting screen time as a feasible public health strategy to improve adolescent self-rated health. Furthermore, it is recommended that schools, families, and communities work together to create a supportive environment that monitors and promotes movement behaviors in adolescents.

### Funding
The authors received no funding for this work.

### Competing Interests
The authors declare there are no competing interests.

### Author Contributions
- Guanghui Shi conceived and designed the experiments, prepared figures and/or tables, and approved the final draft.
- Chaomei Liang conceived and designed the experiments, prepared figures and/or tables, authored or reviewed drafts of the article, and approved the final draft.
- Wanli Zang analyzed the data, authored or reviewed drafts of the article, and approved the final draft.
- Ran Bao performed the experiments, analyzed the data, authored or reviewed drafts of the article, and approved the final draft.

- Jin Yan conceived and designed the experiments, authored or reviewed drafts of the article, and approved the final draft.
- Li Zhou performed the experiments, analyzed the data, authored or reviewed drafts of the article, and approved the final draft.
- Lei Wang analyzed the data, authored or reviewed drafts of the article, and approved the final draft.

## Human Ethics

The following information was supplied relating to ethical approvals (*i.e.,* approving body and any reference numbers):

The study was conducted in accordance with the Declaration of Helsinki and approved by the Institutional Review Board of Shanghai University of Sport, and the grant number is 102772021RT071. The approval date was 24 May 2021.

## Data Availability

The raw measurements are available in the Supplementary File.

## Supplemental Information

Supplemental information for this article can be found online at http://dx.doi.org/10.7717/peerj.16174#supplemental-information.

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
