# Peer review of "24-hour movement behaviours and self-rated health in Chinese adolescents: a questionnaire-based survey in Eastern China"

_PeerJ, doi:10.7717/peerj.16174_

## Round 0.1 · original submission · Minor Revisions

Thanks for submitting your manuscript. The article poses some merits but can be improved by addressing the following issues-

1. It is mentioned that already published articles on the same issue in the Chinese population but why is this study needed? Hence, provide further clarification is needed

2. More information on the data collection process needs to be in the methods section. How were the questionnaires distributed?

3. Page 4 line 93-94, how the participants provided valid data?

4. There needs to be more information on the gatekeeper.

5. How did you deal with the unfilled questionnaire?

6. How was the sample size determined?

7. How the central ethical issues were maintained?

Reviewer 1 ·

Basic reporting

1. To improve reading flow, please write out abbreviations when using them for the first time (lines: 37, 38, 205)

2. Carefully read through line 101 – within my understanding a screen time of “less than 2 hours” stays in line with screen time recommendations.

3. I would suggest using the terms “Male” and “Female” instead of “Boy” and “Girl” in Table 1

4. The language used is clear and professional. However, please read carefully through the paper again focusing on grammar mistakes.

Experimental design

The chosen paper is very interesting and provides relevant knowledge on the topic of “24-hour movement behaviours and self-rated health in adolescents”. It focuses on a holistic health approach by using self-related health as an indicator and does strengthen already published literature in the Chinese area. However, I suggest the following minor revisions:

1. Within the methodology section the recruitment process is not clear enough explained – please provide more in-depth information (e.g., inclusion and exclusion criteria).

2. More evidence is needed why this study uses the sentence “In general, would you say that your health is excellent?” to conduct self-related health (Line 132 – 134).

Validity of the findings

No comment.

Annotated reviews are not available for download in order to protect the identity of reviewers who chose to remain anonymous.

·

Basic reporting

A clear research question is the basis of any effective research effort. A carefully defined research issue must include the identification of study participants, an intervention (or exposure) along with a comparator, and results (Lane,2018). As a first step in formulating a research question, a literature review can provide details on what is already known about a subject, the types of studies that have been carried out, and what questions previous research has left unanswered (Lane,2018). Authors mentioned that prior studies have demonstrated that following the 24-hour movement guidelines had a good influence on physical health. They also identified from previous studies that; 24-hour movement guidelines had a positive impact on mental health also. It is evident that the authors conducted a thorough literature review before beginning this study and successfully found a study gap. From the above discussion, I can say that the issue addressed in this research is clear, concise, and focused. A good research question frequently begins with a basic observation and is then refined iteratively into a well-formulated question using the frameworks PICO, PICo, SPIDER, etc. Aim of this study is clear and identified but the research question and objectives also can be mentioned in the introduction part. Authors can include which formula they have used to formulate their research question for this study. Also they can mention their study objectives in the introduction part.

Experimental design

Gathering data for research is essential since it will help researchers better comprehend a theoretical framework (Bernard,2002). If a convenience sampling technique is used, keep in mind that effectiveness and validity are crucial (Morse,2009). In this study, researchers mentioned about “Convenient” sampling method in line 92. It should be “Convenience” sampling. Convenience sampling involves choosing subjects that are easier to reach. As a result, not all eligible members of the target population will have the same opportunity to participate in the study, and the findings may not necessarily be generalizable to the entire community. There’s no justification mentioned by the authors for using this sampling method. Authors also need to mention inclusion and exclusion criteria for sampling. Pre-validated questionnaire used in this research, ‘Pre-Validated’ term can be mentioned. It is necessary to mention the language of the questionnaire used for the survey and any translations of the questionnaire into other languages if took place (English to Chinese). Participants need to understand the questionnaire properly to respond properly so sometimes translation can be helpful to gather correct responses.

Validity of the findings

According to the study's findings, adolescents' self-rated health is favourably correlated with following the 24-hour movement requirements. The result of the study is precise enough. All supporting data have been offered; they are reliable. Authors performed ordinal logistic regression to identify the association between adherence to the 24-hour movement guidelines and self-rated health. They can mention their dependent and independent variable. All important confounding factors were identified.

Additional comments

This study may help develop health promotion initiatives for adolescents that focus on 24-hour activity behaviour in China. The title might be made more precise by mentioning the study location. According to the authors, this study was conducted in China so it can be titled- ‘24-hour movement behaviours and self-rated health in adolescents in China’

References:
Bernard, H. R. (2002), Research methods in anthropology: Qualitative and quantitative approaches (3rd ed.)’, Walnut Creek, CA: Alta Mira Press.
Lane,S. (2018) ‘A good study starts with a clearly defined question’,BJOG, Vol 125,Issue 9, P 1057.
Morse, J. M., & Niehaus, L. (2009), Mixed method design: Principles and procedures. Walnut Creek, CA: Left Coast Press.

---

## Round 0.2 · accepted · Accept

Thanks for addressing the changes.